# Global Dermatophyte Infections Linked to Human and Animal Health: A Scoping Review

**DOI:** 10.3390/microorganisms13030575

**Published:** 2025-03-03

**Authors:** Aditya K. Gupta, Tong Wang, Mesbah Talukder, Wayne L. Bakotic

**Affiliations:** 1Division of Dermatology, Department of Medicine, Temerty Faculty of Medicine, University of Toronto, Toronto, ON M5S 3H2, Canada; 2Mediprobe Research Inc., London, ON N5X 2P1, Canada; twang@mediproberesearch.com (T.W.); ssusmita@mediproberesearch.com (S.); mesbah.talukder@bracu.ac.bd (M.T.); 3School of Pharmacy, BRAC University, Dhaka 1212, Bangladesh; 4Bako Diagnostics, Alpharetta, GA 30005, USA; wbakotic@bakodx.com

**Keywords:** dermatophyte, ringworm, tinea, zoonoses

## Abstract

Dermatophytes are commonly encountered pathogens in clinical practice causing superficial infections of the skin, hair, and nails. These pathogens are often found on animals such as livestock (e.g., cattle, rabbits) and pets (e.g., cats, hedgehogs) that can lead to spillover infections in human populations. Here, we reviewed published reports (2009–2024) of dermatophyte infections in animals and in humans with a history of animal contact. A literature search was completed in October 2024 using PubMed, Embase (Ovid), and Web of Science (Core Collection), which identified 250 articles. Generally, dermatophytes tend to infect younger animals with long hair and exhibit a species-specific host range. *Microsporum canis* was the most commonly reported species—linked to cats—that can cause tinea capitis, especially concerning the development of kerion in children. *Trichophyton verrucosum* is strongly associated with cattle. The *Trichophyton mentagrophytes* complex shows a diverse range of animal hosts, with rabbits being most frequently reported; however, *T. mentagrophytes* var. *erinacei* is almost exclusively isolated from hedgehogs, and *T. mentagrophytes* var. *benhamiae* is more commonly found on rodents (e.g., guinea pigs). Lastly, the geophilic *Nannizia gypsea* has been isolated from both dogs and cats. Managing dermatophyte zoonoses is an ongoing challenge, as healthcare providers may empirically treat with corticosteroids or antibacterial agents due to its atypical inflammatory appearance. Evidence of in vitro resistance against griseofulvin and fluconazole has been documented in multiple zoonotic dermatophyte species. Resistance development against terbinafine and itraconazole is also a possibility, although the number of reports is scarce. Under the principles of the One Health approach, research on human fungal diseases should take animal and environmental factors into account. A renewed call for increased testing efforts is warranted.

## 1. Introduction

Dermatophyte infections are a common cause of skin diseases in both humans and animals. This group of pathogens was originally suspected to be geophilic but have gradually adapted to infect vertebrates overtime through soil contact and carriage on animal fur [1]. Common dermatophyte species found on animals are often considered to be zoophilic, such as *Microsporum canis* and *Trichophyton* species (*T. verrucosum*, *T. mentagrophytes* var. *mentagrophytes*) [2]. It is suspected that some anthropophilic dermatophyte species can be traced to an animal origin, such as *T. mentagrophytes* var. *interdigitale,* which is suspected to have evolved from zoophilic *T. mentagrophytes* var. *mentagrophytes* often found on rabbits [3]. An animal-to-human transmission may occur via direct contact, such as petting or farming, or indirectly from the environment, as infectious propagules (e.g., arthroconidia) can remain viable for years under optimal temperature and humidity conditions [2].

Unlike animals, humans lack fur, hence an infection by zoophilic or geophilic dermatophytes—outside of their natural habitat—may induce severe inflammations. One example is *M. canis* infections causing tinea capitis in children, which may lead to the secondary development of kerion characterized by a deep follicular invasion, resulting in edematous and pustular lesions with alopecia [4]. Dermatophyte zoonoses is a unique clinical entity that can present a diagnostic challenge due to its inflammatory appearance (e.g., pustules, papules, swelling), which can resemble bacterial infection or noninfectious dermatitis. This increases the risk of mistreatment with one or multiple courses of antibacterial agents or corticosteroids without testing for possible fungal infection. Morrell and Stratman reviewed 51 patients diagnosed with *T. verrucosum* infection—most often linked to a contact history with cattle—in the United States [5], of which 51.0% (26/51) were empirically treated with topical/oral antibiotics or topical corticosteroids. Most patients required specialist referral before being diagnosed and treated for *T. verrucosum* infection, with an average wait time of 41.5 days between the onset of symptoms and the ordering of fungal culture [5]. These findings highlight the importance of taking an animal exposure history, considering occupation (e.g., farmer, breeder), and conducting confirmatory tests for patients presenting with inflammatory lesions atypical for dermatophytoses.

Recently, the U.S. Centers for Disease Control and Prevention (CDC) has advocated for the One Health framework in managing fungal diseases [6], which considers environmental and animal factors in examining the spread and resistance development of human diseases. When a patient is infected with zoophilic dermatophytes, it is advisable to form a collaborative framework between dermatologists and veterinarians to identify the animal source and prevent further spread or re-infection [7]. The lack of antifungal stewardship practices, such as susceptibility testing, is a common issue affecting both human and veterinary healthcare [8]. Abuse of over-the-counter medications, particularly antifungal ointments admixed with corticosteroids, has been linked to the spread of a new dermatophytic species—*T. mentagrophytes* var. *indotineae*—causing severe, recalcitrant dermatophytoses in the Indian subcontinent [8]. Although *T. mentagrophytes* var. *indotineae* has not demonstrated zoonotic potential, healthcare providers are advised to remain vigilant [6] in view of other related species, such as *T. mentagrophytes* var. *erinacei* [9], that can be transmitted from pets.

In this review, we aim to update our current understanding of dermatophytes that impacts both human and animal health. The range of animal hosts and their corresponding dermatophyte species are summarized. Clinical manifestations, treatment challenges, and antifungal susceptibility profiles are also discussed.

## 2. Materials and Methods

A literature search was conducted on 15 October 2024 per Preferred Reporting Items for Systematic reviews and Meta-Analyses (PRISMA) (protocol registration: INPLASY202520036) [10]. The search strategy was developed based on the zoonotic potential of dermatophyte species and their respective animal hosts reported in the literature [2,11]. Three electronic databases were queried: PubMed, Embase (Ovid), and Web of Science (Core Collection). The following subject headings/search terms were used: “*Trichophyton*”, “zoonosis”, “zoonoses”, “*Arthroderma benhamiae*”, “*Arthroderma vanbreuseghemii*”, “*Arthroderma simii*”, “*Microsporum canis*”, “*Nannizzia gypsea*”, “*Microsporum gypseum*”, “bat”, “bird”, “cat”, “calves”, “camel”, “cattle”, “chicken”, “dog”, “equine”, “feline”, “fowl”, “fox”, “goat”, “hedgehog”, “horse”, “livestock”, “leopard”, “llama”, “mammal”, “marmot”, “monkey”, “pet”, “pig”, “porcupine”, “poultry”, “primate”, “ruminant”, “rabbit”, “reptile”, “rodent”, “sheep”, “swine”, “tortoise”, “wildlife”.

De-duplication and title/abstract screening were carried out using Covidence (https://www.covidence.org/). The inclusion criteria were reports of dermatophyte infections in animals, or dermatophyte infections in humans with a reported history of animal contact, published between 2009 and 2024. To differentiate infection from mere colonization, or dermatophyte carriage in animals due to human contamination [2], studies reporting asymptomatic cases or without the reporting of symptoms were excluded. Animal model experiments were excluded. Non-dermatophyte molds and yeasts were excluded. Non-English articles, reviews, conference proceedings, and expert opinions were also excluded.

Data were tabulated using Microsoft Excel (version 2301) with the following parameters: author name, year, region, type of study (animal infection vs. human infections linked to animals), mycology testing, animal type, number of subjects, site of infection, symptom, pathogen identification, antifungal susceptibility testing, treatment. Based on the taxonomic classification proposed by de Hoog et al. in 2017 [12], *Microsporum gypsea* was synonymized with *Nannizia gypsea*; under “one fungus, one name”, the teleomorph *Arthoderma* was synonymized with *Trichophyton* (*A. benhamiae* = *T. benhamiae*, *A. vanbreuseghemii* = *T. mentagrophytes*, *A. simii* = *T. simii*).

## 3. Results and Discussion

Two hundred and fifty articles were identified (Appendix A), involving 22 different animal types presenting with symptomatic dermatophyte infections, or which were linked to symptomatic dermatophyte infections in humans (Figure 1). Of these, the most commonly reported were cats, dogs, cattle, rabbits, rodents, hedgehogs, and horses. Between 2009 and 2024, an increase in the number of published articles reporting animal or zoonotic dermatophytoses was observed beginning in 2018 (Figure 2A). Globally, there were a total of 37 regions reporting human dermatophyte infections linked to animal contact (Figure 2B). The relative distribution of animal hosts corresponding to each of the major dermatophyte species identified is shown in Figure 3.

### 3.1. Microsporum canis (Order: Onygenales; Family: Arthrodermataceae)

Twenty-seven studies reported potential zoonotic *M. canis* infections [13,14,15,16,17,18,19,20,21,22,23,24,25,26,27,28,29,30,31,32,33,34,35,36,37,38,39]. Most cases involved encounters with cats, including pets and strays. Infections linked to contact with symptomatic dogs were also reported [13,14,25]. Sanguansook et al. reported infection of a worker at a cat café—a popular tourist destination in recent years—along with the isolation of *M. canis* in six cats presenting with erythematous lesions with crusts and scales [37]. Two cases of plausible laboratory-acquired infections were reported by Gnat et al. after workers processed specimens from a cat with pruritic, alopecic patches [18], of which one case of kerion developed. Two recent studies from China and Poland confirmed the predominance of *M. canis,* accounting for approximately 80% of cases of zoonotic transmission from cats [15,21]. Furthermore, *M. canis* has demonstrated potential for both zoonotic and human-to-human transmissions. After acquiring an *M. canis* infection from animals, subsequent infections of close contacts (e.g., family members, coworkers) were reported in six studies [18,26,29,30,34,39]. This chain of transmission was verified using molecular techniques such as genotyping and PCR fingerprinting [18,26,34,39].

Clinically, *M. canis* infection commonly causes tinea capitis, characterized by erythematous and pruritic alopecic patches with crusts and scales [16,28,29,30,35,39]. In severe cases, Capoor et al. reported multiple edematous lesions in three children [39]; kerion—characterized by painful, edematous, vesicular, and pustular lesions with alopecia—was also reported in children [23]. Other clinical presentations include tinea faciei [26,30,31,32,34,38], tinea corporis [18,21,25,31,33], tinea manuum [20,34], and onychomycosis [36]. Signs of papules and pustules were also reported in tinea manuum patients who had recently been in contact with infected cats and dogs [20].

*M. canis* infection is a significant cause of morbidity in cats (Figure 3A). Similar of human infections, common symptoms include pruritic, erythematous, focal or multifocal lesions with alopecia [40,41,42,43,44,45,46]. Pseudomycetoma—a rare complication involving deep or subcutaneous fungal invasion associated with a high mortality rate—was reported in six studies [47,48,49,50,51,52]. This condition can be characterized by nodular, ulcerative lesions with yellow granular exudate warranting surgical intervention [47,48]. Histopathological findings include amorphous granules composed of dense, irregular fungal elements surrounded by inflammatory infiltrates (macrophages, neutrophils, lymphocytes) [47,48], with signs of fibrosis [50,52]. A case of “true mycetoma” was described by Kano et al. in a 9-year-old Persian cat, evidenced by the presence of fistulas draining from deep tissues, inflammation, fibrosis, and granules with abundant fungal hyphae [53]. Other symptoms include kerion, miliary dermatitis and folliculitis [54,55]. Risk factors for dermatophytosis in cats include young age (<1 year) [34,40,44,56], longhaired breeds [57], as well as Persian and Scottish Fold breeds [44]. Other less frequently reported animal hosts for *M. canis* include dogs [40,44,56,57,58,59], rabbits [60,61,62], and horses [63,64,65,66].

### 3.2. Nannizia gypsea (Order: Onygenales; Family: Arthrodermataceae)

Although classified as a geophilic dermatophyte, *N. gypsea* has demonstrated the ability to infect animals and humans [11]. It is speculated that animals initially contract *N. gypsea* infections through contact with soil, which can then spread within household environments. Sixteen studies reported potential zoonotic transmissions [15,16,22,27,32,67,68,69,70,71,72,73,74,75,76,77], which predominately involved a contact history with cats and dogs. Romano et al. reported six cases of zoonotic *N. gypsea* infections, including four cases reporting a contact history with symptomatic cats [68]. Clinical presentations include tinea corporis/tinea cruris with erythematous, scaly lesions, and tinea barbae with alopecia, nodules, papules, and pustules [68]. Tobeigei et al. reported evidence of human-to-human transmission involving three children diagnosed with tinea corporis/capitis caused by *N. gypsea*, likely originating from a pet cat, that subsequently led to the infection of three family members, including one child developing tinea capitis with multiple abscesses [70]. A case of tinea corporis in a zoo worker was linked to a porcupine infected with *N. gypsea* presenting with crusty, scaly, macerated, and ulcerative lesions [77].

Two case studies reported tinea capitis in children complicated by the development of kerion [72,74]. The first case presented with painful, edematous, and pustular lesions [74]; DNA sequencing led to the identification *N. gypsea*, which was also found on his dog companion, the indoor carpet, and the doghouse [74]. The second case had diffused alopecia and was linked to a pet guinea pig [72].

Dogs represent the most commonly reported animal host for *N. gypsea* followed by cats (Figure 3B). Predisposing factors for contracting dermatophytosis in dogs include young age (<1–2 years) [56,57], longhaired coats (e.g., Yorkshire Terrier breed [44,57]) [78], living in shelters [78], and signs of kerion or pustular lesions [58]. Besides zoonotic transmission of *N. gypsea* via direct contact, fomites such as furniture, sheets, and carpets can also be the source of infection [79]. Other reported animal hosts for *N. gypsea* in the Global South include horses [63,64,66,80], rabbits [43,62,80], and sheep [81,82], while hedgehogs were reported in the Global North [83,84].

### 3.3. Trichophyton verrucosum (Order: Onygenales; Family: Arthrodermataceae)

Plausible zoonotic infections by *T. verrucosum*—predominately involving cattle and farm workers—were reported in 14 studies [5,15,16,19,22,27,67,73,85,86,87,88,89,90]. Courtellemont et al. reported a higher incidence rate in males, with children being more likely to develop tinea capitis complicated by the development kerion than adults [16]. Among farm workers, tinea corporis was more commonly observed [73,89]. Additionally, the development of kerion associated with tinea barbae, tinea corporis, or tinea capitis was also observed [73,90]. In a study by Łagowski et al., a case of fingernail onychomycosis caused by *T. verrucosum* in a breeder was linked to infected llamas, which was confirmed by PCR fingerprinting [87]. None of the included studies reported evidence of human-to-human transmission.

Cattle living on farms accounted for the majority of the *T. verrucosum* isolates reported in the literature (Figure 3C). Risk factors include young age (e.g., newborn, calf) [91,92,93], intensive or semi-intensive breeding systems [91,93], poor ventilation [91], newly introduced cattle from outside the farm [91], concomitant parasitic infestation [91], infected farm workers [93], and cattle raised for meat production rather than dairy production [91,93]. Similar to humans, infected cattle exhibit symptoms such as erythematous, scaly and crusty lesions with alopecia [42,91,94,95,96]; hair matting, scabbing, and pruritus can also be observed [97,98,99]. Other less commonly reported animal hosts include camels [43,100,101,102], which may present with granulomatous lesions [101,102].

### 3.4. Trichophyton mentagrophytes Complex (Order: Onygenales; Family: Arthrodermataceae)

Taxonomic classification of *T. mentagrophytes* and related species has remained a challenge due to diagnostic ambiguity by conventional methods (e.g., culture), often necessitating molecular diagnosis (e.g., sequencing) for accurate identification. Furthermore, due to incomplete/ongoing speciation, some isolates may require multi-locus sequencing typing for identification [103], which cannot be used readily as part of routine diagnosis. Following the proposal of the “*T. mentagrophytes*-series” by de Hoog et al. [12], Nenoff et al. proposed the term “*T. mentagrophytes* complex” including anthropophilic *T. interdigitale*, as well as zoophilic *T. mentagrophytes*, *T. quinckeanum*, *T. benhamiae*, and *T. erinacei* [104]. Recently, Švarcová et al. proposed a “variety rank” for describing members of the *T. mentagrophytes* complex, such as “*T. mentagrophytes* var. *mentagrophytes*” and “*T. mentagrophytes* var. *interdigitale*” [103].

Diverse infection sites and animal hosts have been observed for the *T. mentagrophytes* complex. Fifty-seven studies reported potential zoonotic infections involving the *T. mentagrophytes* complex [15,16,17,19,21,22,23,27,31,32,67,80,89,105,106,107,108,109,110,111,112,113,114,115,116,117,118,119,120,121,122,123,124,125,126,127,128,129,130,131,132,133,134,135,136,137,138,139,140,141,142,143,144,145,146,147,148], of which most were linked to a contact history with rabbits, hedgehogs, or rodents. Clinical presentations were highly variable, including tinea corporis [129,140], tinea faciei [137,147], tinea manuum [134,143], tinea barbae [118,136], tinea capitis [112,126], and tinea cruris [89,120]. Multi-site involvement is often reported [105,120,128,129,146,148]. Inflammation, pain, vesicles, and pustules [108,110,124,134,138,143], as well as the development of kerion [112,122,126], and Majocchi’s granuloma were also reported [130,148]. Following the initial zoonotic infection, subsequent infections of close contacts were reported in six studies [110,131,132,136,140,147]. Mesquita et al. reported a possible outbreak at a school, where 15 students were infected with *T. mentagrophytes* complex as a result of direct contact with infected rabbits, which led to a subsequent infection of a roommate of one of the infected students [131]. Veraldi et al. reported four children who developed symptoms associated with *T. mentagrophytes* complex infection after contact with pet rabbits [140]. Subsequently, 18 children from the same school contracted *T. mentagrophytes* complex infections [140].

Interestingly, infection by anthropophilic *T. mentagrophytes* var. *interdigitale*—confirmed by multi-locus sequence typing (ITS, D1/D2, β-tubulin)—was linked to a contact history with infected rabbits in four patients [144]. The appearance of a scalp infection (kerion) in one patient suggests a zoonotic origin [144].

Despite an unclear species delineation based on phylogeny, the ecological niche for the *T. mentagrophytes* complex is diverse (Figure 3D). Rabbits accounted for the majority of the animal hosts reported in the literature followed by dogs and hedgehogs. Epidemiological surveys have identified the predominance of the *T. mentagrophytes* complex in farm rabbits compared to *M. canis* or *N. gypsea* [61,62,149]. Predisposing factors include old age, high temperature, high humidity, and multifocal lesions [61,62,149]. Symptoms include alopecic patches with erythema and scaling [43]; inflammation and folliculitis can also be observed [149]. Other animal hosts include dogs, which may also exhibit inflammatory symptoms [43,58]. Hedgehogs were predominately associated with *T. mentagrophytes* var. *erinacei* infection (see Section 3.4.1).

#### 3.4.1. *Trichophyton mentagrophytes* var. *erinacei*

Based on studies reporting animal dermatophyte infections, *T. mentagrophytes* var. *erinacei* is almost exclusively isolated from hedgehogs [57,83,84,150,151], in contrast to the rest of the *T. mentagrophytes* complex (Figure 3E). Often kept as exotic pets, the two most common species are the African pygmy hedgehog (*Atelerix albiventris*) and the European hedgehog (*Erinaceus europaeus*). Eighteen studies reported potential zoonotic infections [32,107,109,111,116,117,121,123,124,127,133,134,135,136,139,141,143,145]. Of note are severe cases of tinea manuum, characterized by pustular, pruritic lesions with pain [109,117,124,134,141,143,145]; bullous eruptions, erosive inflammation, and fever were also reported [109,124,134,143].

Among wild European hedgehogs, Le Barzic et al. reported that up to 23.3% (96/412) of isolates were *T. mentagrophytes* var. *erinacei*, which includes animals presenting with erythematous scaly and crusty lesions with spine loss, as well as asymptomatic carriers [83]. Similar findings were reported by Gnat et al. confirming European hedgehogs as a reservoir, with reduced in vitro susceptibility to griseofulvin and fluconazole [84]. *T. mentagrophytes* complex isolates from wild European hedgehogs also demonstrated resistance against terbinafine conferred by mutations in the squalene epoxidase gene (*SQLE*) [84]. Furthermore, a survey of European hedgehogs across 10 European countries and New Zealand identified *T. mentagrophytes* var. *erinacei* isolates with the ability to produce penicillin-like antibiotics, which raises the possibility of selecting the methicillin-resistant *Staphylococcus aureus* (MRSA) phenotype in case of concomitant bacterial colonization/infection [152].

#### 3.4.2. *Trichophyton mentagrophytes* var. *benhamiae*

Potential zoonotic infections by *T. mentagrophytes* var. *benhamiae*—mostly involving guinea pigs—were reported in 12 studies [22,105,106,108,115,119,122,128,132,137,138,142]. Inflammatory symptoms, such as swelling, papules, pustules, and skin erosion, were reported in patients with tinea faciei and tinea corporis [108,138,142]. A case of severe tinea capitis with kerion was reported in a child linked to a contact history with a symptomatic pet guinea pig [122].

Reported animal hosts for *T. mentagrophytes* var. *benhamiae* were predominately rodents (guinea pigs, chinchillas) (Figure 3F). Peano et al. reported signs of desquamation, crusting, inflammation, follicular casts, and alopecia in guinea pigs [153]. Signs of hyperkeratosis with alopecic patches were also reported in porcupines and alpacas [154,155], as well as lymphocyte infiltration and follicular epithelial hyperplasia per histopathologic examination [154].

### 3.5. Other Trichophyton spp.

Less frequently reported *Trichophyton* spp. implicated in zoonotic infections include *T. tonsurans*, *T. violaceum*, *T. equinum*, and *T. simii*. After being bitten by a dog on the forearm, Zheng et al. reported a case of Majocchi’s granuloma caused by *T. tonsurans* [156], characterized by a nodular, pustular lesion with deep lymphocyte and neutrophil infiltration and hyperplasia of the stratum spinosum. An Egyptian survey identified *T. violaceum* as the dominant pathogen in symptomatic dermatophytosis patients (37.2% [60/161]), of which 32.8% (20/60) reported a contact history with pets [19]. Two cases of tinea cruris and tinea capitis caused by *T. equinum* were linked to contact with horses [157,158], including one study reporting contact with horses that had previously exhibited dermatophytosis symptoms [158]. Another study reported two cases of tinea corporis linked to an asymptomatic dog detected with *T. equinum* [159]. An unusual case of otomycosis caused by *T. simii* was reported after the patient’s ear was touched by a monkey [160].

### 3.6. Diagnostic Challenge of Dermatophyte Zoonoses

The risk of misdiagnosing dermatophytoses at the point-of-care has been recognized in the literature, and confirmatory testing is recommended [161]. With dermatophyte zoonoses, atypical inflammatory symptoms can mislead healthcare providers to empirically prescribe corticosteroids (e.g., clobetasone), antibacterial (e.g., doxycycline) or antiviral agents (e.g., acyclovir), and allergy medications (e.g., loratadine), potentially causing disease exacerbation. Twenty-six case studies reporting misdiagnosed cases of zoonotic dermatophyte infections are summarized in Table 1.

The majority of reported cases were linked to infections by the *T. mentagrophytes* complex. Mazur et al. reported a case of an immunocompetent, young adult with a lower leg infection—linked to a pet guinea pig—complicated by the development of Majocchi’s granuloma [130]. The lesions were characterized by erythema, desquamation, pruritus, papules, and pustules around hair follicles, which had progressed to resemble furuncles [130]. Mistreatment with topical corticosteroids and antibiotics possibly led to the development of nodular, inflammatory lesions, as well as the subsequent infection of a family member [130]. Zhang et al. reported a pediatric case with tinea corporis and tinea capitis, linked to infected farm rabbits with lesions and alopecia [148]. The patient had no co-morbidities; initial presentation included infiltrating erythema, scaling, crusting, alopecia, and pruritus, which worsened with the development of papulopustules and abscesses due to self-treatment with dexamethasone cream [148]. Sidwell et al. reported a case of kerion tinea barbae caused by *T. mentagrophytes* var. *erinacei*—linked to a pet hedgehog—characterized by multiple, coalescing lesions with pustules [136]. Initial presentation led to a misdiagnosis of impetigo, and the patient was administered oral and intravenous antibiotics [136]. Subsequently, a close contact contracted *T. mentagrophytes* var. *erinacei* infection through osculatory transfer [136].

In a case series, Starace et al. reported eight cases of pediatric tinea incognito caused by *N. gypsea* that were previously misdiagnosed as eczema and treated with corticosteroids [69]. In a case report, a 3-year-old child presenting with painful, pruritic, crusting and purulent lesions with alopecia and abscesses was diagnosed with *T. verrucosum* infection, which likely originated from cattle on the family farm [86]. Initial misdiagnosis and mistreatment by a general practitioner led to disease worsening with the development of a hypersensitivity reaction (i.e., dermatophytid) due to the use of the potent topical corticosteroid clobetasone [86].

Drug-induced eruptions are considered uncommon occurrences with terbinafine or itraconazole treatments [162,163], and their causes remain unclear. In a case report, an elderly onychomycosis patient treated with terbinafine (250 mg/d), with concomitant medications including prednisone, doxazosin mesylate, and aspirin, developed macular eruptions and cervical lymphadenopathy after 4.5 weeks [163]. The reaction resolved in 6 weeks after the discontinuation of terbinafine and was considered idiosyncratic, as the patient has been taking prednisone for several months prior with no adverse reactions [163]. The differential diagnosis includes dermatophytid reactions, which can be suspected based on the development of intensely pruritic lesions distant from the site of infection without detectable fungal elements [164].

From the included studies (Table 1), four patients exhibited paradoxical reactions to terbinafine or itraconazole [26,85,122,138]. In a case of tinea corporis, the adult patient presented with a lesion of the nape caused by *T. mentagrophytes* var. *benhamiae*, which was initially mistreated with topical oxytetracycline-hydrocortisone ointment, then oral tetracycline with topical sedum fusidate [138]. The subsequent administration of terbinafine (125 mg/d) led to the development of pruritic rashes of the trunk and limbs, which was deemed a dermatophytid reaction that did not warrant the discontinuation of terbinafine [138]. In contrast, another immunocompetent adult patient developed drug-induced exanthema after receiving terbinafine 250 mg/d for 2 weeks, warranting a switch to itraconazole 200 mg/d [85]. The patient presented with tinea corporis caused by *T. verrucosum*, which subsequently progressed into a deep follicular infection after being empirically treated with oral amoxicillin-clavulanic acid and oral ciprofloxacin [85]. In another case of kerion tinea capitis in a child (*T. mentagrophytes*), after receiving empirical oral clarithromycin treatment, a presumed drug-induced eruption occurred with terbinafine treatment (62.5 mg/d) that led to the switch to griseofulvin [122]. Authors noted in retrospect that a dermatophytid reaction should have been considered [122]. A similar report described a pediatric patient with tinea corporis/faciei caused by *M. canis*, previously treated with topical steroids, who developed skin eruptions with itraconazole treatment (dosage unspecified) and subsequently switched to terbinafine [26]. In two studies, oral glycyrrhizin (100 mg/d twice daily for 2–4 weeks) was tried to prevent the development of hypersensitivity reactions [86,148].

### 3.7. Antifungal Resistance (In Vitro and Clinical Resistance)

Antifungal susceptibility testing (AFST) results per the broth microdilution method (CLSI, Clinical and Laboratory Standards Institute; EUCAST, European Committee on Antimicrobial Susceptibility Testing) are summarized in Table 2. Among the standard antifungals, all of the identified dermatophyte species exhibited high minimum inhibitory concentrations (MICs) against griseofulvin and fluconazole, including animal isolates. Clinical resistance to griseofulvin was reported in cats infected with *M. canis* [55]. A tinea manuum patient infected with *T. mentagrophytes* var. *erinacei* was unresponsive to topical clotrimazole and oral griseofulvin 500 mg/d after one week [117]. Another patient with kerion tinea capitis caused by *T. mentagrophytes* var. *benhamiae* had positive culture after receiving griseofulvin 10 mg/kg/d for 6 months [122]. In five cases of *N. gypsea* infections causing tinea capitis, oral fluconazole treatment was not effective [70,74].

Selected studies have reported high terbinafine MICs (≥1 µg/mL) in *M. canis* [24,65,165], the *T. mentagrophytes* complex [84], and *T. equinum* [65], albeit with limited reports on clinical resistance. Interestingly, *T. mentagrophytes* var. *indotineae*, a newly emerged anthropophilic clonal offshoot of *T. mentagrophytes* var. *mentagrophytes* associated with recalcitrant dermatophytosis globally, has been reported in a stray dog with high MICs against terbinafine as well as griseofulvin, ketoconazole, and itraconazole [168]. An Indian study by Thakur et al. reported whole-genome sequencing results of *T. mentagrophytes* var. *interdigitale* and *T. mentagrophytes* var. *indotineae* isolates of both human and animal origins [169]. Although none of the animal isolates demonstrated in vitro resistance against terbinafine, in contrast to human isolates, the results of the phylogenetic reconstruction—demonstrating close relatedness (<200 SNPs)—suggest zoonotic transmissions [169]. In a cat infected with *M. canis*, topical terbinafine 1% applied daily for 3 month did not lead to improvement, and subsequent AFST found a high terbinafine MIC of ≥32 µg/mL [165]. Another study reported a child with tinea capitis caused by *M. canis*; initial treatment with oral terbinafine 80 mg/d and topical naftifine/ketoconazole cream was not effective after 4 weeks (albeit with a terbinafine MIC of 0.03 µg/mL), which led to the switch to itraconazole 100 mg/d for 7 weeks, resulting in clinical and mycological cure [30]. In a Polish survey of wild European hedgehogs, Gnat et al. reported two *T. mentagrophytes* var. *mentagrophytes* isolates that grew on solid media supplemented with 1 µg/mL of terbinafine, with corresponding *SQLE* mutations [84].

High itraconazole MICs (≥0.5 µg/mL) were reported in *M. canis* [24,30,65,166], *N. gypsea* [166], *T. verrucosum* [65,86,91], and the *T. mentagrophytes* complex [65,84,91,137,166,168]. In an immunocompromised patient with tinea faciei and onychomycosis—linked to a contact history with a hedgehog—caused by *T. mentagrophytes* var. *erinacei*, oral itraconazole (4 months; unspecified dosage) and fluconazole (6 months; unspecified dosage) were not effective [133]. Subsequently, the patient was treated successfully using terbinafine for 4 months (unspecified dosage) [133]. Clinical resistance to itraconazole has been reported in cats infected with *M. canis* [48,53,55]. In a case of feline pseudomycetoma caused by *M. canis* with concomitant *Staphylococcus aureus* colonization, a 4-week course of oral itraconazole 10 mg/kg/d, cephalexin 20 mg/kg twice daily, and topical ketoconazole was not effective [48]. Subsequent administrations of intralesional amphotericin B and oral terbinafine led to a partial response, but large nodular lesions remained [48]. In a similar case, itraconazole 10 mg/kg twice daily was not effective after 5 weeks [53]. Subsequent up-dosing of itraconazole to 30 mg/kg twice daily for 3 months was also ineffective [53].

There is scarce information on the use of ketoconazole, as well as the third-generation triazoles (voriconazole, posaconazole). Topical ketoconazole applied for 8 weeks was not effective in treating a case of tinea corporis caused by *T. mentagrophytes* var. *benhamiae* transmitted from a pet guinea pig; the patient later responded to a 3-week course of oral terbinafine 125 mg/d [132]. Another case of tinea capitis caused by *M. canis* unresponsive to topical ketoconazole was successfully treated using oral itraconazole 3 mg/kg/d for 6 weeks [29]. A Chinese study reported voriconazole and posaconazole MICs of 0.6 µg/mL in *T. mentagrophytes* var. *benhamiae* isolated from a child with tina faciei [137]. However, none of the included studies reported evidence of clinical resistance.

## 4. Conclusions

This review is limited by potential publication bias and restriction of the literature search to English-language publications, which may affect data representativeness. We could not exclude the possibility that certain geographical regions may be overrepresented in the literature. Nonetheless, to our knowledge, our work represents the first comprehensive review of dermatophyte infections impacting both human and animal health, with detailed analyses on the animal host range, antifungal resistance, and clinical challenges related to the lack of diagnostic testing and mistreatment. Available data suggest that selected dermatophyte species have demonstrated zoonotic potential, thereby causing infections of varying severity; this can lead to the emergence of new pathogens. Under the framework of One Health, a holistic approach should be considered for the management of dermatophytosis, as these pathogens—especially concerning drug resistant strains—are present in the environment and on animals. A better understanding of zoophilic dermatophytes linked to spillover infections—in terms of animal reservoir, species spectrum, mode of transmission, clinical presentation, and antifungal susceptibility profile—would better inform future surveillance efforts and help devise mitigation strategies. Confirmatory testing and antifungal susceptibility testing remain essential in these efforts, and continued advocacy is warranted to expand access and encourage uptake.

## Figures and Tables

**Figure 1 microorganisms-13-00575-f001:**
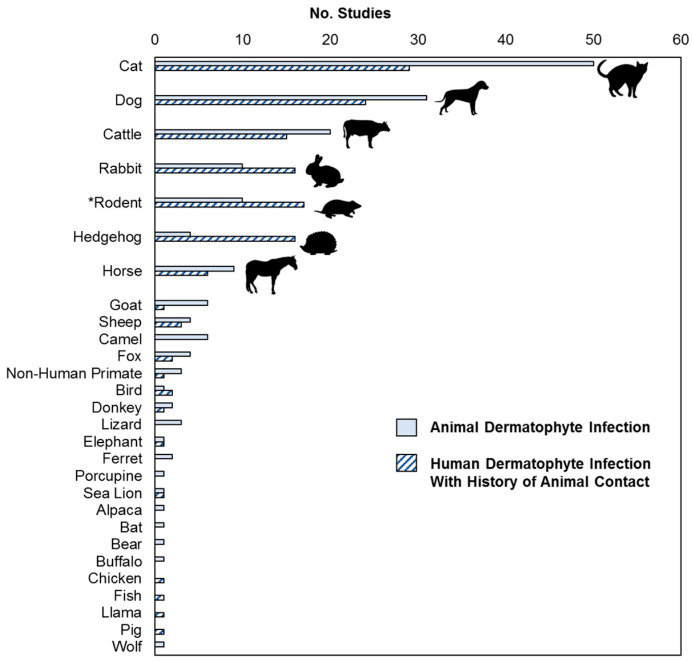
Distribution of animal types reported across the included studies. * Inclusive of chinchilla, degus, guinea pig, hamster, mice, rat, and squirrel.

**Figure 2 microorganisms-13-00575-f002:**
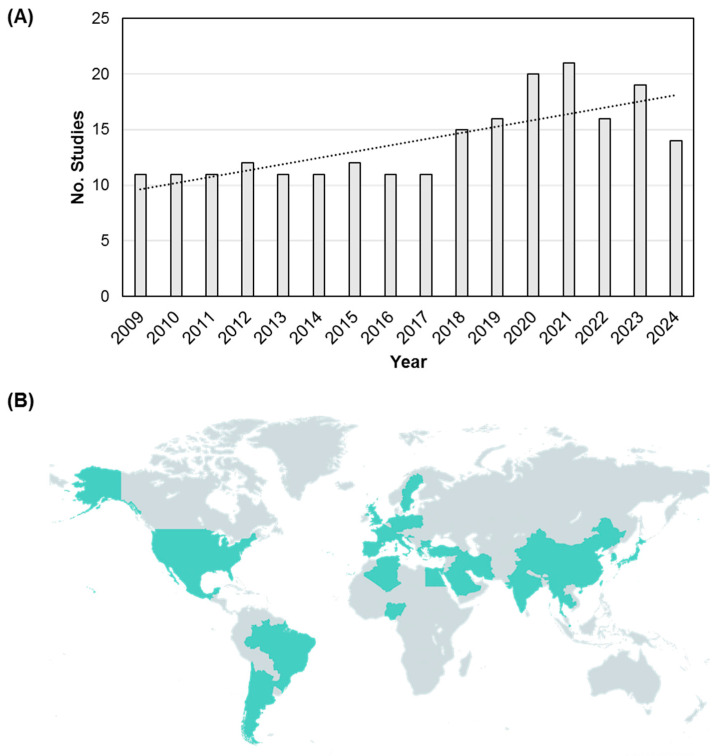
(**A**) Number of included studies stratified per the publication year. (**B**) Global regions reporting human dermatophyte infections with a history of animal contact.

**Figure 3 microorganisms-13-00575-f003:**
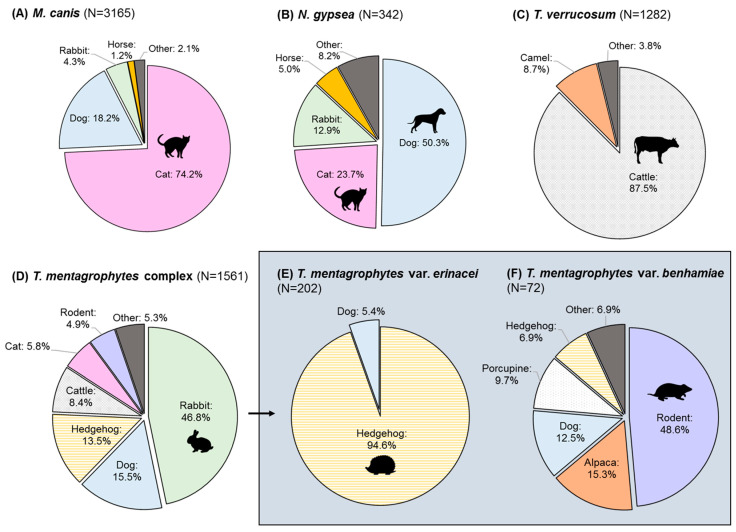
Relative distribution of animal hosts reported for (**A**) *M. canis*, (**B**) *N. gypsea*, (**C**) *T. verrucosum*, (**D**) *T. mentagrophytes* complex, (**E**) *T. mentagrophytes* var. *erinacei*, and (**F**) *T. mentagrophytes* var. *benhamiae*. *T. tonsurans*, *T. violaceum*, *T. simii* and *T. equinum* are not shown due to limited reporting.

**Table 1 microorganisms-13-00575-t001:** Summary of reported difficult-to-diagnose or mistreated cases of dermatophyte zoonoses.

Symptoms	Regimen	Outcome
*M. canis*		
Pediatric tinea capitis (N = 1) [35]: pruritus, erythema, alopecia	Oral antibiotics and anti-allergy medications × 2 weeks	Symptoms became severe with the appearance of erythema and alopecia
	Itraconazole 50 mg/d × 1 month	Improved
Pediatric tinea corporis (N = 1) [26]: erythema, scales, pruritus	Topical steroid ointments	Lesion spread
	Oral itraconazole + topical ketoconazole	**Drug eruption**
	Oral terbinafine	Resolved
Pediatric tinea faciei, corporis (N = 1) [38]: erythema, scales, pruritus	Loratadine, diphenhydramine, fluocinolone acetonide for presumed allergy	Worsened
	Acyclovir 600 mg/d × 3 days for presumed viral infection	Nausea, vomiting, headache
	Terbinafine 250 mg/d × 1 month	Resolved
** *N. gypsea* **		
Pediatric tinea capitis with kerion (N = 1) [74]: swelling, pain, pustular, alopecia	Topical hydrocortisone butyrate, ciclopirox, amikacin creams applied daily	No improvement
	Added oral fluconazole × 2 weeks	Lesion became swollen, painful, and suppurating
	Oral griseofulvin 25 mg/kg/d × 8 weeks + topical isoconazole and diflucortolone dressings then topical terbinafine 1% dressings applied daily	Resolved
Pediatric tinea capitis with kerion (N = 1) [72]: pustules, swelling, alopecia, fever, abscesses	Oral cefixime 50 mg/kg/d + topical mupirocin ointment twice daily × 4 days	No improvement
	Itraconazole 150 mg/kg/d × 8 weeks then oral prednisolone 30 mg/d + topical terbinafine twice daily	Resolved
Pediatric tinea corporis, faciei (N = 8) [69]: erythema, scales, perifollicular casts	Topical corticosteroids for presumed eczema	Minimal improvement
	Topical ciclopirox olamine 1% twice daily × 3 weeks	Resolved
** *T. verrucosum* **		
Adult tinea corporis (N = 1) [85]: erythema, swelling	Surgical incision; oral amoxicillin-clavulanic acid; oral ciprofloxacin	Further swelling, pruritus, new satellite lesions
	Surgical incision; terbinafine 250 mg/d × 2 weeks	Symptoms improved; **drug eruption**
	Itraconazole 200 mg/d	Resolved
Pediatric tinea capitis, corporis (N = 1) [86]: erythema, scales, abscesses, alopecia, crusting, purulent discharge	Topical clobetasone butyrate cream	**Dermatophytid**; modified lesion appearance to maculopapular, strong pruritus, lymph node swelling without fever
	Itraconazole 100 mg/d × 10 weeks + glycyrrhizin 200 mg/d × 2 weeks + topical butenafine hydrochloride cream	Resolved
***T. mentagrophytes*** **complex**		
Adult tinea barbae (N = 1) [136]: purulent crusted lesion	Oral clarithromycin for presumed impetigo	No improvement
	Intravenous clindamycin and acyclovir (changed to oral after 2 days) + potassium permanganate soak daily	NR
	Added itraconazole 200 mg/d × 6 weeks and oral ciprofloxacin × 3 weeks	Resolved
Adult tinea barbae (N = 1) [118]: papules, pustules, swelling	Topical and oral antibiotics × 2 weeks	NR
	Itraconazole 100 mg/d × 2 months	Relapsed
	Itraconazole 100 mg/d × 2 months	Resolved
Adult tinea corporis (N = 1) [138]: acne-like papules, scales, circumscribed erythema	Oxytetracycline 30 mg/g; hydrocortisone alcohol 10 mg/g	Enlarged papules
	Oral tetracycline; topical sodium fusidate	No improvement
	Terbinafine 125 mg/d × 2 months	**Dermatophytid**; treatment continued and symptoms improved
Adult tinea corporis (N = 1) [107]: highly pruritic, erythema	Topical gentamicin-betamethasone dipropionate	Minimal improvement of pruritus
	Itraconazole 200mg/d × 10 days	Improved
Adult tinea corporis (N = 1) [123]: erythema, pruritus, seropapules, scales, crusts	Topical corticosteroid	Enlarged lesion
	Topical corticosteroid continued for presumed contact dermatitis × 2 weeks	No improvement
	Itraconazole pulse + topical luliconazole × 7 days	Improved
Adult tinea corporis with Majocchi’s granuloma (N = 1) [130]: pustules, papules, nodules, erythema, desquamation, pruritus	Topical glucocorticosteroids + antibacterial agents	Relapsed with increased severity
	Oral antibiotics	No improvement
	Oral terbinafine + topical isoconazole, mazipredone with miconazole and econazole × 6 weeks	Resolved
Adult tinea corporis, cruris (N = 1) [120]: erythema, papules, plaques	Oral doxycycline	No improvement
	Methylprednisolone 32/16 mg/d × 6 months for presumed disseminated eczema	No improvement
	Itraconazole 200 mg/d × 14 days then 400 mg/d × 7 days + topical antifungal-glucocorticosteroids	Improved
Adult tinea manuum (N = 1) [141]: pruritus, bullae, pain	Oral corticosteroid taper + topical antibiotic	NR
	Oral doxycycline	Increasing pain and tense bullae
	Oral terbinafine + topical econazole × 4 weeks	Resolved
Adult tinea manuum (N = 1) [109]: pustule, pain, pruritus, fever	Oral Augmentin + topical betamethasone dipropionate + topical gentamicin + topical miconazole + potassium permanganate compresses for presumed contact dermatitis with secondary pyoderma	New pustules developed
	Oral terbinafine × 2 weeks	Resolved
Pediatric tinea capitis with kerion (N = 1) [122]: swelling, desquamation, pain, fever	Oral clarithromycin 15 mg/kg/d + topical miconazole + topical terbinafine	Enlarged lesion
	Terbinafine 62.5 mg/d × 7 days	**Drug eruption**
	Griseofulvin 10 mg/kg × 6 months	Relapsed
	Griseofulvin 10 mg/kg × 2 months	Improved
Pediatric tinea capitis, corporis with Majocchi’s granuloma (N = 1) [148]: erythema, scales, alopecia, pruritus, papules, pustules, abscesses	Topical dexamethasone acetate cream twice daily	Enlarged lesions
	Oral itraconazole 100 mg/d × 12 weeks + oral glycyrrhizin 100 mg/d × 4 weeks + topical butenafine hydrochloride 1% daily	Resolved
Pediatric tinea capitis, faciei, corporis (N = 1) [147]: scale, pustule, inflammation, alopecia, and subcutaneous nodules on the scalp; erythema on the face and trunk	Debridement, topical and oral antibacterial treatment for presumed impetigo	Lesions became severe; fever and chills
	Itraconazole 100 mg/d + topical ketoconazole 2% shampoo + povidone iodine solution × 10 days Intravenous ceftriaxone sodium 500 mg/d + intravenous dexamethasone 7.5 mg/d × 6 days	Resolved
Pediatric tinea corporis (N = 1) [114]: papules, seropapules, pustules, erythrosquamous lesion	Topical ointment for presumed eczema	No improvement
	Topical isoconazole 1%/diflucortolone valerate 0.1% twice daily × 2 weeks	Resolved
Pediatric tinea faciei (N = 1) [137]: pruritus, erythematous, annular plaque	Topical clobetasol propionate/ketoconazole cream × 15 days	No improvement
	Topical pimecrolimus/hydrocortisone butyrate cream	Lesion became tender, pruritic, and transformed into a “ring” shape
	Terbinafine 125 mg/d + topical sertaconazole nitrate cream twice daily × 4 weeks	Improved
Pediatric tinea faciei (N = 1) [135]: pruritus, erythema, scales	Topical corticosteroids	No improvement
	Itraconazole + topical betamethasone-clotrimazole	Worsened after cessation of itraconazole
	Terbinafine + topical clotrimazole	Resolved
Pediatric tinea faciei, corporis (N = 1) [113]: erythema, scales, pruritus	Topical cortisone + antibiotic for presumed microbial eczema	Relapsed with severe inflammation
	Terbinafine 125 mg/d × 5 weeks + topical isoconazole 1%-diflucortolone valerate 0.1% × 10 days then topical ciclopirox	Improved
Pediatric tinea manuum (N = 1) [134]: pruritic, pustules, erythema, web space maceration	Topical steroids × 4 weeks	Lesion spread
	Itraconazole 200 mg/d + topical isoconazole/diflucortolone valerate cream × 4 weeks	Resolved
** *T. tonsurans* **		
Adult tinea corporis with Majocchi’s granuloma (N = 1) [156]: swelling, nodules, pain, pruritus, pustular, scales	Oral cephalosporin	No improvement
	Itraconazole 200 mg/d; moxibustion on the swollen area	Resolved

Regimens are sorted based on chronological order; mistreatments without obtaining fungal testing results are highlighted. NR, not reported.

**Table 2 microorganisms-13-00575-t002:** Antifungal susceptibility testing results of human and animal dermatophyte isolates.

Country (Year)	Antifungal	MIC Range (µg/mL)	AFST	Reference
*M. canis*				
China (2018)	Terbinafine	>32	CLSI M38-A2	[165]
	Itraconazole	0.023		
China (2013)	Terbinafine	0.03	CLSI M38-A2	[30]
	Ketoconazole	2		
	Itraconazole	0.5		
Egypt (2017)	Griseofulvin	1	CLSI M38-A2	[65]
	Terbinafine	1		
	Fluconazole	32		
	Itraconazole	0.5		
Greece (2010)	Griseofulvin	0.064–8	CLSI M38-A2	[24]
	Terbinafine	0.032–4		
	Fluconazole	0.25–64		
	Itraconazole	0.064–1		
	Posaconazole	0.032–0.5		
India (2024)	Griseofulvin	4	CLSI M27A4	[39]
	Terbinafine	0.06		
	Itraconazole	0.125		
Iran (2021)	Griseofulvin	0.064–2	CLSI M38-A2	[166]
	Terbinafine	0.016–0.064		
	Ketoconazole	0.016–0.064		
	Fluconazole	0.25–4		
	Itraconazole	0.002–0.5		
Poland (2022)	Griseofulvin	0.25	CLSI M38Ed3	[84]
	Terbinafine	0.016		
	Ketoconazole	0.125		
	Fluconazole	16		
	Itraconazole	0.125		
	Voriconazole	0.064		
** *N. gypsea* **				
India (2019)	Griseofulvin	16	CLSI M38-A2	[71]
	Terbinafine	0.0156		
	Itraconazole	0.0625		
Iran (2021)	Griseofulvin	0.064–2	CLSI M38-A2	[166]
	Terbinafine	0.016–0.064		
	Ketoconazole	0.016–0.064		
	Fluconazole	0.25–4		
	Itraconazole	0.002–0.5		
Poland (2022)	Griseofulvin	0.5	CLSI M38Ed3	[84]
	Terbinafine	0.0125		
	Ketoconazole	0.125		
	Fluconazole	8		
	Itraconazole	0.25		
	Voriconazole	0.016		
** *T. verrucosum* **				
China (2019)	Terbinafine	0.004	CLSI M38-A2	[86]
	Itraconazole	0.5		
	Voriconazole	0.0625		
Egypt (2020)	Griseofulvin	0.5–4	CLSI M38-A2	[91]
	Terbinafine	0.03–0.25		
	Fluconazole	16–64		
	Itraconazole	1–4		
Egypt (2017)	Griseofulvin	0.5	CLSI M38-A2	[65]
	Terbinafine	0.5		
	Fluconazole	16		
	Itraconazole	1		
***T. mentagrophytes*** **complex**				
China (2019) *	Terbinafine	0.0315	CLSI	[148]
	Itraconazole	0.125		
	Voriconazole	0.0625		
	Posaconazole	0.0625		
China (2018) ^†^	Terbinafine	0.015	CLSI M38-A2	[137]
	Fluconazole	4		
	Itraconazole	1		
	Voriconazole	0.6		
	Posaconazole	0.6		
Czech Republic (2021) ^‡^	Terbinafine	0.004–0.016	EUCAST E.Def 11.0	[129]
	Ketoconazole	0.016–1		
	Fluconazole	2–64		
	Itraconazole	0.008–0.125		
	Efinaconazole	0.008–0.064		
Egypt (2020) *	Griseofulvin	0.25–2	CLSI M38-A2	[91]
	Terbinafine	0.06–0.5		
	Fluconazole	8–32		
	Itraconazole	0.25–1		
Egypt (2017) *	Griseofulvin	0.5	CLSI M38-A2	[65]
	Terbinafine	0.03		
	Fluconazole	8		
	Itraconazole	1		
Iran (2024) *	Griseofulvin	0.5–16	CLSI M38-A3	[167]
	Terbinafine	0.016		
	Ketoconazole	0.016–2		
	Itraconazole	0.016–0.125		
Iran (2024) ^§^	Griseofulvin	≥16	CLSI M38-A3	[168]
	Terbinafine	≥16		
	Ketoconazole	≥16		
	Itraconazole	≥16		
Iran (2021) *	Griseofulvin	0.128–2	CLSI M38-A2	[166]
	Terbinafine	0.016–0.125		
	Ketoconazole	0.016–0.128		
	Fluconazole	0.5–4		
	Itraconazole	0.002–1		
Iran (2020) ^†^	Griseofulvin	1	CLSI M38-A2	[105]
	Terbinafine	0.063		
	Ketoconazole	1		
	Itraconazole	0.25		
	Voriconazole	0.125		
	Posaconazole	0.063		
Poland (2022) *	Griseofulvin	1	CLSI M38Ed3	[84]
	Terbinafine	(growth on terbinafine agar)		
	Ketoconazole	0.5		
	Fluconazole	32		
	Itraconazole	0.5		
	Voriconazole	0.016		
Poland (2022) ^†^	Griseofulvin	0.5	CLSI M38Ed3	[84]
	Terbinafine	0.008		
	Ketoconazole	0.25		
	Fluconazole	16		
	Itraconazole	0.125		
	Voriconazole	0.032		
Poland (2022) ^¶^	Griseofulvin	1	CLSI M38Ed3	[84]
	Terbinafine	0.004		
	Ketoconazole	0.5		
	Fluconazole	16		
	Itraconazole	0.25		
	Voriconazole	0.032		
Poland (2018) ^‡^	Terbinafine	0.004–0.016	CLSI M38Ed3	[125]
	Ketoconazole	0.125–1		
	Fluconazole	2–32		
	Itraconazole	0.03–0.25		
	Voriconazole	0.03–0.25		
** *T. equinum* **				
Egypt (2017)	Griseofulvin	1	CLSI M38-A2	[65]
	Terbinafine	1		
	Fluconazole	16		
	Itraconazole	0.25		
Poland (2018)	Griseofulvin	0.125–0.25	CLSI M38Ed3	[159]
	Terbinafine	0.125–0.5		
	Ketoconazole	0.125–0.5		

* *T. mentagrophytes* var. *mentagrophytes* identified based on ITS sequencing. ^†^ *T. mentagrophytes* var. *benhamiae* identified based on ITS sequencing. ^‡^ *T. mentagrophytes* var. *quinckeanum* identified based on ITS sequencing. ^¶^ *T. mentagrophytes* var. *erinacei* identified based on ITS sequencing. ^§^ *T. mentagrophytes* var. *indotineae* identified based on ITS sequencing.

## Data Availability

The raw data supporting the conclusions of this article will be made available by the authors on request.

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
