# Peer review of "Global Dermatophyte Infections Linked to Human and Animal Health: A Scoping Review"

_microorganisms, 2025, doi:10.3390/microorganisms13030575_

Round 1
Reviewer 1 Report
Comments and Suggestions for Authors
The paper has certain value in terms of research topics and content. However, there are still some aspects that can be improved. The following are the specific review
Q1: It is recommended that the author conduct a more in - depth analysis of the causes of misdiagnosis.
Q2: The author should further collect relevant data to improve the analysis of antifungal drug resistance mechanisms.
Q3: It is recommended to further explore how factors such as the living habits and immune status of animal hosts affect the infection and transmission of fungi, and analyze the causes and potential risks of carrying specific fungi to better understand the transmission mechanism of zoonotic fungal diseases.
Q4: The article mentions the One Health approach, but the elaboration in the content is not in-depth.
Q5: It is recommended that the author further highlight the differences and unique contributions of this study compared with previous studies in the discussion section, such as elaborating on the comprehensiveness of data collection and the depth of analysis.
Author Response
The paper has certain value in terms of research topics and content. However, there are still some aspects that can be improved. The following are the specific review
Authors: Thank you for taking the time to review our work.
Q1: It is recommended that the author conduct a more in - depth analysis of the causes of misdiagnosis.
Authors: Thank you for your suggestion. We agree that misdiagnosis is an important issue and have added a new subsection (3.5.) and table (Table 1). Here, we have summarized reported clinical cases that initially mislead healthcare providers into prescribing antibacterial/antiviral agents, corticosteroids, or allergy medications, and treatment outcomes.
Q2: The author should further collect relevant data to improve the analysis of antifungal drug resistance mechanisms.
Authors: Thank you for your comment. We agree on the importance of deciphering the mechanism of resistance. However, based on studies where antifungal susceptibility testing was conducted (see Table 2), only one study examined mutations of the squalene epoxidase gene conferring terbinafine resistance (Gnat et al. 2022 doi:10.1007/s00248-021-01866-w). Reviews of antifungal resistance mechanisms have been published (Martinez-Rossi et al. 2021. doi: 10.3390/jof7080629; Gupta et al. 2025. doi: 10.1371/journal.ppat.1012913).
Q3: It is recommended to further explore how factors such as the living habits and immune status of animal hosts affect the infection and transmission of fungi, and analyze the causes and potential risks of carrying specific fungi to better understand the transmission mechanism of zoonotic fungal diseases.
Authors: Thank you for your comment. Based on large-scale surveillance studies of animals that we identified in this review, the immune status has not been tested or reported as a significant risk factor (Allizond et al. 2015. doi: 10.1007/5584_2015_5004; Lopes et al. 2024. doi: 10.3390/microorganisms12081727; Seker et al. 2011 doi: 10.1016/j.prevetmed.2010.11.003; Long et al. 2020 doi: 10.1136/vr.105957). In farm animals, intensive breeding systems, poor ventilation, and humidity can contribute to the development of dermatophyte infections (lines 231-236; 295-296). Based on clinical presentations, it can be presumed that animal-to-human transmission occurs via direct contact, such as with tinea manuum from hedgehogs, which are often kept as exotic pets. The reason behind the host selectivity for specific dermatophyte species identified in this study is unclear. One could speculate that this selectivity may be due to differences in enzyme profiles or geographical restrictions. Further studies are warranted to explain the underlying mechanisms.
Q4: The article mentions the One Health approach, but the elaboration in the content is not in-depth.
Authors: We agree with the reviewer on the importance of providing actionable strategies under the One Health framework. In the manuscript, we have provided an example of forming a collaboration between physicians and veterinarians (lines 68-70), which has been successfully tried in one study (Cruciani et al. 2021 doi: 10.3389/fvets.2021.718766.). At present, we need more advocacy, as well as expanding testing capacities, before local or state regulations can be established based on this principle. Another focal point for One Health is “dual-use” antifungal agents in human healthcare and agriculture (e.g., treatment naïve patients contracting azole-resistant Aspergillus infections). However, we believe this point is beyond the scope of this review and have already been described in detail (Fisher et a. 2024. doi: 10.1038/s44259-024-00055-2).
Q5: It is recommended that the author further highlight the differences and unique contributions of this study compared with previous studies in the discussion section, such as elaborating on the comprehensiveness of data collection and the depth of analysis.
Authors: Thank you for your comment. We have revised the Conclusions section to highlight the contribution of our work to the literature (lines 479-482).
Reviewer 2 Report
Comments and Suggestions for Authors
The manuscript presents a scoping review of global dermatophyte infections affecting both human and animal populations, with an emphasis on zoonotic transmission. It provides a comprehensive literature review spanning 2009–2024, focusing on various dermatophyte species and their respective hosts, clinical manifestations, treatment challenges, and antifungal resistance patterns. The authors effectively frame the study within the One Health approach, advocating for interdisciplinary collaboration in managing dermatophytoses.
I suggest the following major revisions:
- The manuscript acknowledges potential publication bias and the exclusion of non-English studies, which could lead to regional disparities in reported data.
- The discussion on the taxonomy of the Trichophyton mentagrophytes complex remains ambiguous, with varying nomenclatures used throughout the text. A more standardized taxonomic framework would enhance clarity.
- While the study compiles a vast dataset, some sections lack critical analysis. The discussion could benefit from more direct comparisons between species in terms of transmission patterns, clinical severity, and antifungal susceptibility.
- The manuscript reports high MIC values for certain antifungals, particularly griseofulvin and fluconazole, but does not extensively discuss clinical implications or alternative treatment strategies.
- The manuscript includes several figures summarizing host distribution and infection trends, but their resolution and clarity could be improved for better readability.
- Typographical and Formatting Issues: Several minor grammatical errors and inconsistencies in reference formatting should be addressed.
- Redundant Information: Some sections, particularly within the results, contain repetitive descriptions of clinical presentations across different dermatophyte species. Condensing these sections would improve readability.
Author Response
The manuscript presents a scoping review of global dermatophyte infections affecting both human and animal populations, with an emphasis on zoonotic transmission. It provides a comprehensive literature review spanning 2009–2024, focusing on various dermatophyte species and their respective hosts, clinical manifestations, treatment challenges, and antifungal resistance patterns. The authors effectively frame the study within the One Health approach, advocating for interdisciplinary collaboration in managing dermatophytoses.
I suggest the following major revisions:
- The manuscript acknowledges potential publication bias and the exclusion of non-English studies, which could lead to regional disparities in reported data.
Authors: Thank you for taking the time to review our work. We have stated this as a study limitation under Conclusions (lines 476-478).
- The discussion on the taxonomy of the Trichophyton mentagrophytes complex remains ambiguous, with varying nomenclatures used throughout the text. A more standardized taxonomic framework would enhance clarity.
Authors: Thank you for your comment. For members of the T. mentagrophytes complex, we have revised the nomenclatures throughout the text for consistency (e.g., T. mentagrophytes var. interdigitale, T. mentagrophytes var. erinacei).
- While the study compiles a vast dataset, some sections lack critical analysis. The discussion could benefit from more direct comparisons between species in terms of transmission patterns, clinical severity, and antifungal susceptibility.
Authors: Thank you for your suggestions. In this review, we have presented the relative distribution of animal hosts reported for M. canis, N. gypsea, T. verrucosum and the T. mentagrophytes complex (Figure 3). Direct contact with these companion or farm animals will increase the risk of transmission. We agree that further critical analyses would improve the utility of our work, however we are limited by the fact that most of the included studies are case reports. In terms of clinical severity, while we see a higher number of reports on inflammatory lesions attributed to T. mentagrophytes complex infections, we could not exclude the possibility of reporting bias, and are limited by the lack of mechanistic studies to explain this trend. Furthermore, severe symptoms such as kerion in pediatric tinea capitis patients have also been found in infections caused by M. canis, N. gypsea and T. verrucosum. Similarly, we found this dataset unsuitable for an in-depth comparison of antifungal susceptibility profiles between species due to inconsistencies in the number of reports.
- The manuscript reports high MIC values for certain antifungals, particularly griseofulvin and fluconazole, but does not extensively discuss clinical implications or alternative treatment strategies.
Authors: We agree with reviewer that detailed management strategies for antifungal resistance would improve the utility of our work. However, reports of clinical resistance/treatment failure was scarce based on our review of the current literature, and is mostly limited to isolated case reports. Hence, we could not present comprehensive strategies for tackling this growing issue. These reports have been summarized under Section 3.6.
We have added a section (3.5.) to highlight the implications of empirical treatments without obtaining fungal testing results. We hope this work will encourage more research to optimize the management of dermatophyte zoonoses.
- The manuscript includes several figures summarizing host distribution and infection trends, but their resolution and clarity could be improved for better readability.
Authors: Thank you for your suggestion. We have enlarged Figures 2-3 for better readability.
Comments on the Quality of English Language
- Typographical and Formatting Issues: Several minor grammatical errors and inconsistencies in reference formatting should be addressed.
Authors: Thank you for taking the time to review our work. We have checked our manuscript for grammatical errors and reference formatting errors.
- Redundant Information: Some sections, particularly within the results, contain repetitive descriptions of clinical presentations across different dermatophyte species. Condensing these sections would improve readability.
Authors: Thank you for your comments. We have checked and removed redundant passages about common clinical presentations/symptoms in the Results and Discussion section.
Reviewer 3 Report
Comments and Suggestions for Authors
The submitted manuscript is well written and covers interesting topics, the epidemiology of infections caused by dermatophytes and host dependence is convincingly presented. I do not have major comments. Please remove the wording 'soil-loving', 'animal-loving' and 'human-loving' species from the introduction section, as the definition of 'geophilic', 'zoophilic' and 'anthropophilic' species is sufficient. Furthermore, I recommend that in the case of a described fungal species, its classification and systematic nomenclature be briefly presented in the manuscript text after the first mention.
Author Response
The submitted manuscript is well written and covers interesting topics, the epidemiology of infections caused by dermatophytes and host dependence is convincingly presented. I do not have major comments. Please remove the wording 'soil-loving', 'animal-loving' and 'human-loving' species from the introduction section, as the definition of 'geophilic', 'zoophilic' and 'anthropophilic' species is sufficient.
Authors: Thank you for taking the time to review our work. We have removed the wording from the Introduction section (line 37, 40, 42).
Furthermore, I recommend that in the case of a described fungal species, its classification and systematic nomenclature be briefly presented in the manuscript text after the first mention.
Authors: Thank you for your comment. We have added classification and systematic nomenclature in the respective section headings (lines 129-130, 175, 208-209, 240-241).
Round 2
Reviewer 1 Report
Comments and Suggestions for Authors
The quality of the manuscript has been improved after the revision. But some minor writing issues still need to be fixed.
Author Response
The quality of the manuscript has been improved after the revision. But some minor writing issues still need to be fixed.
Authors: Thank you for taking the time to review our work. We have thoroughly reviewed the manuscript for inconsistencies and typographical errors.